# Differences in morphology, mitochondrial genomes, and reproductive compatibility between two clades of parasitic wasps *Aphelinus mali* (Hymenoptera: Aphelindae) in China

**Min Su[1,2], Lingjun Du[1], Muhammad Yasir Ali[3,4], Jianing Yu[1], Mengyu Chi[1], Ziwen Teng[1], Yinjun Fan[1], Xiumei Tan[1], Hongxu Zhou[1] \***

1 College of Botanical Medicine, Shandong Engineering Research Center for Environment-Friendly Agricultural Pest Management, China-Australia Joint Institute of Agricultural and Environmental Health, Qingdao Agricultural University, Qingdao, China, 2 Agricultural and Rural Bureau of Zhangqiu District, Jinan, China, 3 MARA-CABI Joint Laboratory for Bio-Safety, Institute of PlantProtection, Chinese Academy of Agricultural Sciences, Beijing, China, 4 Key Laboratory of Insect Ecology and Molecular Biology, College of Plant Health and Medicine, Qingdao Agricultural University, Qingdao, China

\* hxzhouqd@sina.cn

**Data Availability Statement:** All relevant data are within the manuscript and its Supporting Information files.

## Abstract

*Aphelinus mali* (Haldeman) (Hymenoptera: Aphelinidae) in China is comprised of two clades (termed, the Shandong and Liaoning clades). In order to clarify the genetic relationship between these two clades, we compared and analyzed the morphological characteristics and the mitochondrial genome of each, and performed a hybridization experiment. Morphological results showed that both males and females of the Liaoning clade were larger than Shandong clade, in terms of whole body, abdominal, wing and antennal lengths, however, there were no significant differences between clades for total length of the middle or hind leg of females. The length of the mitochondrial genome of the Shandong clade was 14415 bp and, for the Liaoning clade, it was 14804 bp. Each contained 31 genes, including 13 protein-encoded genes, 16 tRNA genes, and 2 rRNA genes. The highest AT level among the 13 protein-coding genes for the two clades were the same gene (*ATP8*) (Shandong clade, 91.52%; Liaoning clade, 90.91%). By hybridization and backcrossing, we found that there was no cross incompatibility between these two clades of *A. mali*. Our results indicate that the historic geographical isolation between these clades has not yet caused reproductive isolation of these populations, and they belong to the same species.

## Introduction

Woolly apple aphid (WAA), *Eriosoma lanigerum* (Hausmann) (Hemiptera: Aphididae), is an important quarantine pest of apples in many parts of the world [1]. In recent years, the damage caused by this aphid has become a serious problem in China [2]. Surveys in Rizhao, Shandong

**Funding:** The author reports the following source of funding: National Natural Science Foundation (31371994) awarded to ZH. the funder, Taishan Mountain Scholar Constructive Engineering Foundation of Shandong, China, has played a role in study design, data collection, and preparation of the manuscript.

**Competing interests:** The authors have declared that no competing interests exist.

Province from 2000 to 2002, found that in about 8000 hectares of orchard, with 10–20% of the trees being infested by WAA, causing an annual loss of $5 \times 10^6$ kg of apples, and the pest range has continued to increase [3, 4]. The endoparasitoid *Aphelinus mali* (Haldeman) (Hymenoptera: Aphelinidae) is the dominant natural enemy of *E. lanigerum* in China [5]and is considered the most effective biocontrol agent of this aphid [6, 7]. *Aphelinus mali* has been introduced into 51 countries and has established in 42 of them [8].

*Aphelinus mali* was introduced into China twice: the first introduction was from Japan into Dalian and Lvshun, Liaoning Province (122˚31'E, 39.20˚N) in 1942, while the second was from the former Soviet Union in 1950 into Qingdao, Shandong Province (116˚41'E, 39˚91'N) [9]. High intra-specific variability is expected when a species has a large geographic distribution, as does *A. mali* [10]. Based on the study of the mitochondrial COI gene of this parasitoid, *A. mali* in China is comprised of two regional clades (named the Shandong and Liaoning clades) (Fig 1) [11, 12]. The *A. mali* population in Qingdao and Tai'an in Shandong province and the population in Dalian and Huludao in Liaoning province differ in their biological characteristics and biological control potential [13, 14].

The study of the genetic relationship between these two clades can help to assess the ecological adaptability of each clade and determine whether they have undergone intra-specific differentiation. Such information can provide new ideas and methods for the introduction, reproduction, and field release of *A. mali* as a biological control agent against *E. lanigerum*. The morphological characteristics and mitochondrial genome were compared to clarify the genetic relationships between two clades of *A. mali*. The experimenters also performed a hybridization experiment to see if any reproduction isolation has developed between these clades.

## Materials and methods

### Morphological comparison of clades

To obtain parasitoids for morphological comparisons, *E. lanigerum* aphids were collected from apple orchards in Tai'an, Shandong province (117˚13E, 36˚19N) and Dalian, Liaoning province (121˚52E, 38˚95N), in mid-August, 2014 (Fig 1) [11]. Parasitized aphids, as determined by their blackened and mummified appearances, were noted and placed in Petri dishes and held at 25˚C for parasitoid emergence. Emerged adults of *A. mali* were removed daily, placed in 100% ethanol, and preserved at 20˚C. Male and female *A. mali* were separated based on the larger body size of females, antennal shape (the third section of the female antenna at the funicular joint is square, while that of the male is rectangular) [15], and the abdominal shape (females have a short and thick abdomen, while in male it is slenderical) [9].

Images of body regions or parts were taken with the universal video imaging system (LY-WN-HPCCD (10),Chengdu Li Yang Precision Mechanical and Electrical Co., Ltd., China). We measured the length of the body and abdomen, the length and width of front and hind wings, the length of the whole leg, tibia, tarsus, anterior tarsal segment, and calcar (spine at the tip of the tibia) of the anterior, middle, and hind legs, the length and width of the rod section of the antenna, and the first, the second, and the third antennal segments. One hundred adults of *A. mali* (50M and 50F) from each sample area were measured.

### Mitochondrial genome sequencing

For mitochondrial sequencing, aphids parasitized by *A. mali* were collected from one abandoned apple orchard in Tai'an, Shandong Province and another in Huludao, Liaoning province (120˚87E, 40˚77N), in May 2015. At each site, the five-point sampling method was used, sampling ten trees at each of the five points. On each of these 50 trees per location, five branches bearing blackened *E. lanigerum* aphids that were parasitized by *A. mali* were collected and put

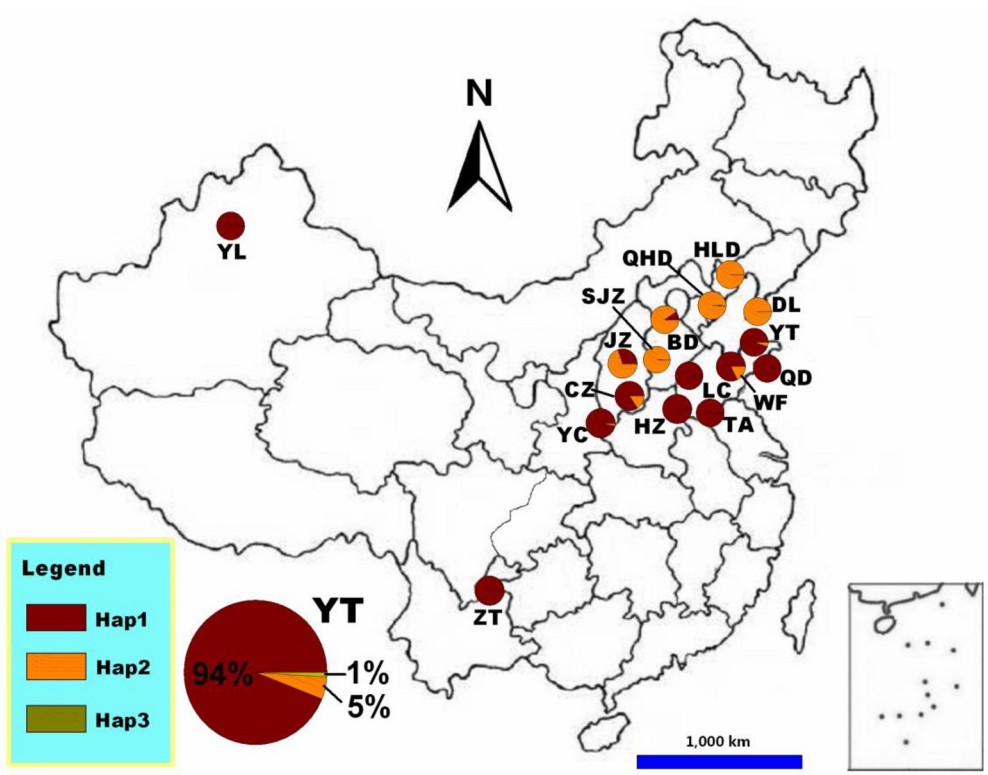

**Fig 1. Sampling sites and haplotype frequencies in the 16 populations of *A. mali* [11].**

into a 25°C incubator in the laboratory to await until *A. mali* emergence. One hundred adults of *A. mali* were obtained from each site and placed individually in 1.5 ml centrifuge tubes containing absolute ethanol and sent to Wuhan Transduction Bio Co., LTD for sequencing.

Primers designed to match generally conserved regions of target mtDNA were used to amplify short fragments from *COX3*, *ATP6*, *COX1*, *NAD5*, *NAD4*, *CYTB*, *NAD1*, *RRNL*, *RRNS* (S1 Table). Specific primers were designed based on these conserved regions sequences and used to amplify the remained mtDNA sequence in several PCR reactions. Primers were designed to produce amplicons with overlaps of about 100 bp. The PCR reaction was performed using the LA Taq polymerase. The PCR products were sequenced directly, or if needed first cloned into a pMD18-T vector (Takara, JAP) and then sequenced, by the dideoxynucleotide procedure, using an ABI 3730 automatic sequencer (Sanger sequencing) using the same set of primers. All obtained fragments were quality-proofed (electropherogram) and BLASTed to confirm that the amplicon was the actual target sequence. Whenever the quality was suboptimal, sequencing was repeated. All obtained fragments were BLASTed to confirm that the amplicon was the target sequence. Mitogenome was assembled stepwise with the help of DNAstar v7.1 program, making sure that the overlaps were identical, and that no NUMTS were incorporated into the sequence.

## Cross-mating experiment

For the cross-mating experiment, *A. mali* as parasitized aphids on apple trees (as described below) were collected from Qingdao (120°39E, 36°31N) to represent the Shandong clade and from Qinhuangdao (119°48E, 39°83N) for the Liaoning clade (Fig 1) [11].

The apple orchard with less management was chosen and the apple trees that had colonies of *E. lanigerum* have been protected with screen cages (1m×1m×2m). Thereafter, the parasitized black aphids were removed daily until they no longer appeared to ensure there were no *A. mali* in the *E. lanigerum* colony, and then kept it in its natural state in the apple orchard, waiting for parasitization by the *A. mali*.

Crosses and reciprocal crosses between the two populations (one per clade) were made for four groups: Liaoning♀×Shandong♂, Liaoning♂×Shandong♀ as the experimental groups, and Shandong♀×Shandong♂, Liaoning♂×Liaoning♀ as control groups; all crosses were repeated 15–20 times. The *A. mali* were kept in the Petri dishes (13.5 cm diameter, had cotton ball with 10% honey water) at 25˚C, 70% RH, and a 16:8 h L:D photoperiod. For each cross, we recorded the number of hosts attacked by each mated, crossed female and the number of offspring, and these offspring's sex ratio (as % female).

The F1 generation of each hybrid cross were backcrossed with the parents of each population, conducting a "within hybrid (F1) population" cross as the control group, with 3–5 replicates per type of crossing. We recorded the number of parasitoid offspring for each crossed female, for all crossing combinations, as well as the mean sex ratio (MSR) of the resulting backcrossed progeny (as % female). The relative compatibility of an inter-group crossing (A × B) was expressed as [16, 17]:

$$\frac{\text{MSR}(A♀ \times B♂)}{\text{MSR}(A♀ \times B♂)} \times 100\%$$

## Statistical analysis

Morphological measurements as well as the hybridization and backcross rates are presented as means ± standard deviations (SD), calculated using Statistical Product and Service Solutions (SPSS) 19.0. Significant differences were determined using one-way analysis of variance (ANOVA) corrected by SPSS 19.0. The independent samples *t*-test was used to analyze the data on parthenogenesis between two clades, using SPSS 19.0.

For genomic analysis, we used MITOS Webserver (http://mitos.bioinf.uni-leipzig.de/index.py) to annotate and analyze the mitochondrial genome sequence and obtain the linear alignment of the mitochondrial genome sequence. The AT-skew and GC-skew were calculated using DNAS-TAR as follows:

$$AT - \text{skew} = \frac{(A\% - T\%)}{(A\% + T\%)}$$

$$GC - \text{skew} = \frac{(G\% - C\%)}{(G\% + C\%)}$$

## Results

### Comparison of morphological characters

Measurements of body dimensions (Table 1) showed that both males and females from the Liaoning clade were bigger than those of the Shandong clade (Fig 2), in terms of the whole body, abdominal, wing, and antennal segment lengths. But the total length of the middle and hind legs showed no significant differences in females between two clades.

### Mitochondrial genome sequencing and analysis

The mitochondrial genome had 14415 bp for the Shandong clade and 14804 bp for Liaoning. Both clades contained 31 genes, including 13 protein-encoded genes, 16 tRNA genes, and 2

**Table 1. Morphological measurements of two clades of *Aphelinus mali* in China.**

| Character length | | Females (Mean ± SD) | | Males (Mean ± SD) | |
|---|---|---|---|---|---|
| | | Shandong | Liaoning | Shandong | Liaoning |
| Whole body | | 810.93 ± 16.03b | 870.29 ± 12.18a | 679.34 ± 12.34d | 758.38 ± 15.95c |
| Abdomen | | 384.12 ± 9.75b | 424.43 ± 7.87a | 292.96 ± 7.53d | 339.94 ± 8.04c |
| Fore wing | | 682.40 ± 11.09c | 770.86 ± 9.39a | 609.84 ± 8.42d | 735.96 ± 8.35b |
| Hind wing | | 303.46 ± 4.99c | 336.84 ± 4.14a | 277.70 ± 4.02d | 321.52 ± 3.51b |
| Fore leg | Femur | 176.70 ± 3.37c | 195.25 ± 2.52a | 167.46 ± 2.14d | 188.30 ± 2.24b |
| | Tibia | 147.32 ± 2.67b | 163.43 ± 2.29a | 138.67 ± 2.20c | 159.98 ± 2.36a |
| | Tarsus | 158.73 ± 2.87b | 168.58 ± 2.87a | 150.56 ± 2.18c | 166.04 ± 2.48a |
| | Pre-tarsus | 19.36 ± 0.41b | 24.14 ± 1.36a | 19.54 ± 0.46b | 23.31 ± 0.38a |
| | Spur | 26.23 ± 0.96bc | 27.56 ± 0.56b | 25.72 ± 0.71c | 29.17 ± 0.76a |
| | Total leg | 528.32 ± 8.42b | 551.39 ± 6.96a | 476.22 ± 6.33c | 543.33 ± 9.01a |
| Middle leg | Femora | 214.75 ± 3.14c | 238.31 ± 3.11a | 200.52 ± 3.22d | 228.08 ± 2.85b |
| | Tibia | 249.91 ± 4.53c | 276.08 ± 3.51a | 229.60 ± 3.50d | 265.51 ± 3.05b |
| | Tarsus | 202.98 ± 3.74b | 215.30 ± 3.24a | 191.74 ± 2.87c | 210.43 ± 3.03a |
| | Pre-tarsus | 19.36 ± 0.41b | 23.26 ± 0.29a | 19.87 ± 0.42b | 22.82 ± 0.29a |
| | Spur | 66.91 ± 1.67b | 74.05 ± 1.15a | 66.27 ± 1.40b | 72.19 ± 2.87a |
| | Total leg | 753.51 ± 12.23a | 752.95 ± 9.44a | 641.73 ± 9.09c | 728.55 ± 10.62b |
| Hind leg | Femora | 253.72 ± 4.56a | 256.15 ± 3.75a | 223.63 ± 3.44c | 239.85 ± 2.87b |
| | Tibia | 259.67 ± 4.06c | 288.79 ± 4.02a | 231.29 ± 3.68d | 274.67 ± 3.75b |
| | Tarsus | 226.86 ± 4.23b | 242.70 ± 3.72a | 211.99 ± 3.41c | 233.09 ± 3.30b |
| | Pretarsus | 19.79 ± 0.42b | 23.30 ± 0.34a | 19.64 ± 0.44b | 23.24 ± 0.32a |
| | Spur | 32.20 ± 1.16c | 36.54 ± 0.73a | 28.60 ± 0.92d | 34.16 ± 0.68b |
| | Total leg | 792.33 ± 12.32a | 810.94 ± 11.57a | 686.55 ± 9.61c | 768.52 ± 12.10b |
| Antenna | Funicle1 | 16.52 ± 0.44b | 19.03 ± 0.31a | 16.27 ± 0.43b | 18.96 ± 0.32a |
| | Funicle2 | 18.04 ± 0.30b | 19.33 ± 0.30a | 16.44 ± 0.29c | 18.94 ± 0.30a |
| | Funicle3 | 33.12 ± 0.44c | 36.57 ± 0.54b | 46.83 ± 0.76a | 45.18 ± 1.52a |
| | Club | 91.13 ± 1.37d | 106.76 ± 1.03b | 100.10 ± 2.19c | 113.90 ± 1.72a |
| | Total antenna | 158.81 ± 2.18c | 181.68 ± 1.68b | 179.62 ± 2.92b | 212.50 ± 2.55a |

Data are mean ± SD (μm), and different letters in the same row indicate significant differences.

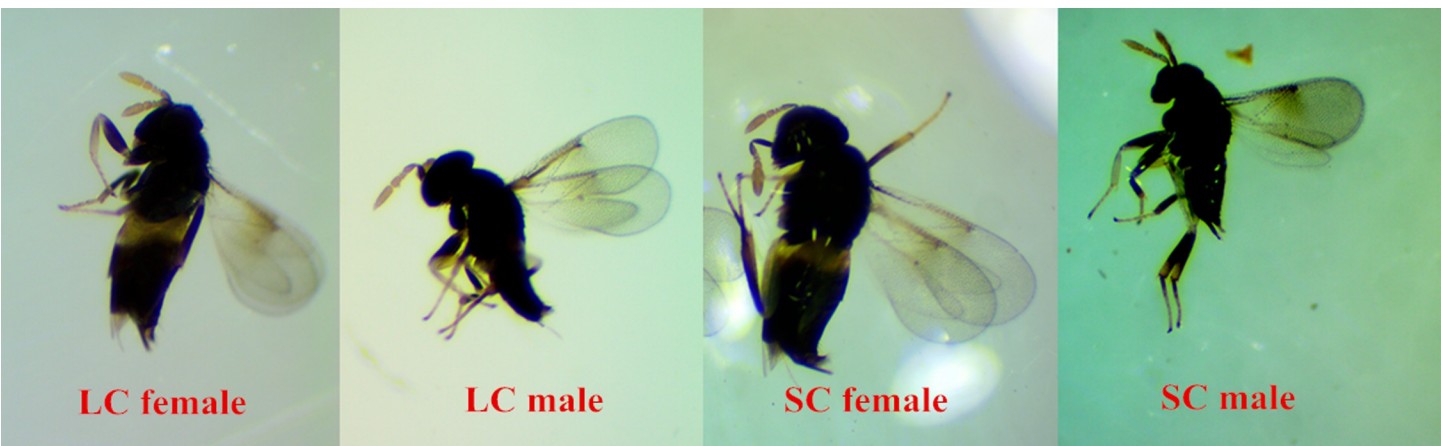

**Fig 2. The images of male and female from two clades of *A. mali*.** (Note: LC: Liaoning Clade; SC: Shandong Clade).

rRNA genes (S2 Table and S1 Fig). The unmeasured area was in the A+T-rich region, which usually contains six tRNA genes (*TRNY, TRNM, TRNI, TRNQ, TRNW*, and *TRNS1*) in such genomes, but which were absent in *A. mali*. The A+T content of both clades were within the range of the Hymenoptera, while the AT skew were higher than other Hymenoptera, with rearrangements of the tRNA and protein-coding genes. The A+T content in the measured region was 84.44% of the Shandong clade and 83.99% of the Liaoning clade, the AT skew was 0.1291 (Shandong clade) and 0.1305 (Liaoning clade), and the GC skew was -0.1619 (Shandong clade) and -0.1328 (Liaoning clade) (S3 Table). For comparison, other Hymenoptera have the A+T content range from 76–86% (S3 Table) [18].

Among the 13 protein-coding genes, the gene with the highest AT content was the *ATP8* gene, in both clades, with an AT content of 91.52% for the Shandong clade and 90.91% for the Liaoning clade. Also, in both clades, the protein-coding gene with the lowest AT content was the cox1 gene, being 77.41% for the Shandong clade and 77.09% for the Liaoning clade (S4 Table).

There were two ribosomal RNA genes, *RRNL* and *RRNS*. The lengths of the *RRNL* gene were 1359bp (in the Shandong clade) versus 1360bp (Liaoning clade), and for these ribosomal genes, the AT content was 87.97% for the Shandong clade and 88.03% for the Liaoning clade, and the *RRNL* gene occurred in an area between the *TRNL1* gene and the *TRNA* gene. The *RRNS* gene had 769bp (Shandong clade) compared to 771bp for the Liaoning clade, and the AT content of the *RRNS* gene was 88.82% in the Shandong clade and 88.98% in the Liaoning clade, and this gene occurred in an area between the *TRNA* gene and the *TRNV* gene.

The length of the tRNA genes of the two clades ranged from 59bp to 75bp in the mitochondrial genome. Three of the 16 tRNA genes, *TRNS2, TRNR* and *TRNV*, could not form the normal clover-shaped secondary structure, unlike the other 13 tRNA genes which could do so in both clades.

## Hybridization experiment

In both clades, unmated *A.mali* females could parasitize *E. lanigerum* aphids and all offsprings were males, indicating neither clade was parthenogenetic.

Crosses between the two clades of *A. mali* led to the production of fertile offsprings (F1 generation). There were no significant differences in emergence rates or sex ratio (% F) among progeny between both clades (Table 2).The relative compatibility of the crosses was greater than 0.75 in both crossing directions (L♂×S♀: 1.188, S♂×L♀: 0.964).

In backcrosses, there were no significant differences in the emergence rates or sex ratio (% F) of progeny from backcrosses between parental clades and the hybrids from crossings of two clades. The relative compatibility values of the two clades were all greater than 0.75 (Table 3).

**Table 2. Results of crosses between the two *Aphelinus mali* clades, in terms of the number of pupae, progeny produced, emergence rate, female ratio and relative compatibility of hybridization.**

|  | # pupae | # adult progeny | % emergence | Sex ratio (%F) | relative compatibility |
|---|---|---|---|---|---|
| L♂×S♀ (n = 18) | 5.28 ± 2.93a | 4.67 ± 3.03ab | 85.09± 16.75a | 0.57 ± 0.28a | 1.188 |
| S♂×L♀ (n = 15) | 4.86 ± 4.28a | 3.78 ± 3.62a | 77.94 ± 28.87a | 0.53 ± 0.32a | 0.964 |
| S♂×S♀ (n = 15) | 7.00 ± 4.95a | 6.37 ± 1.69ab | 91.00 ± 13.98a | 0.48 ± 0.33a | —— |
| L♂×L♀ (n = 18) | 8.67 ± 2.89a | 7.29 ± 2.39b | 84.08 ± 30.29a | 0.55 ± 0.34a | —— |
|  | df = 3; $F = 1.147$ $P = 0.343$ | df = 3; $F = 2.058$ $P = 0.123$ | df = 3; $F = 1.076$ $P = 0.372$ | df = 3; $F = 0.133$ $P = 0.940$ | —— |

Data in the same column are means ± SD, different letters indicate significant differences ($P<0.05$). L: Liaoning clade. S: Shandong clade. n represents the number of replications.

## Discussion

We found differences in some morphological characteristics between the two clades of *A. mali* present in China. Because there were 6 tRNAs that were not detected in both clades, the mitochondrial genomes general structure of the two clades could not be determined and compared. Also, the secondary structures of the *TRNS2*, *TRNR*, *TRNV* genes were incomplete, and there were two tRNA mismatches between clades. Crosses between the two clades produced fertile F1 females, showing that there were no reproductive barriers between these two geographical populations, and that they have not evolved into two species.

Within species, the body size, developmental duration, and diapause responses of insects frequently vary in different geographical regions [19–21]. The body size of animals often changes along gradients of latitude or altitude (Bergmann's rule and the Converse and Bergmann's rule) [22]. Bergmann's rule postulates that body size of mammals and insects increases with greater latitude [21–23], perhaps photoperiod causes influences [19]. So, we speculate that the observed difference in body size between the two *A. mali* clades may be due to their occurrence in locations of different latitude.

Mitochondrial genome sequences are an important marker in molecular systematics that are often used in phylogenetic studies to resolve unclear evolutionary relationships among closely related insects [24–26]. The AT and GC skew values reflect deviation from a population's base composition, which has important inference value for studying the mechanism of mitochondrial genome replication and transcription [27]. Skew values (AT: Shandong clade 0.1291, Liaoning clade 0.1305; GC: Shandong clade -0.1619, Liaoning clade -0.1328) of the *A. mali* clades were normal and in the general range for wasps [18]. The A+T content of other Hymenoptera is ranged about 76% to 86% [18], and the values of these two clades of *A. mali* (Shandong clade: 84.44%, Liaoning clade: 83.99%) were within this range. The present technology cannot determine the complete sequence of the mitochondrial genome of some insects, including *A. mali* [28], however, the genetic measures we obtained suggest that the two clades of *A. mali* belonged to the same species.

Crossing experiments are commonly used to resolve species relationships in the parasitic Hymenoptera order because laboratory cultures of these wasps are often maintained during biological control projects [10]. According to Pinto et al. [16], relative reproductive compatibility index values which are below 0.75 suggest partial reproductive isolation among populations [19, 20], since the productive compatibility index values we obtained for the two *A. mali* clades (L♂×S♀: 1.188, S♂×L♀: 0.964) were far above 0.75 value, so we conclude that these

**Table 3. The number of pupae, progeny produced, emergence rate, sex ratio (as %F)and relative compatibility of backcross.**

| | # pupae | # progeny | % emergence | Sexratio (% F) | relative compatibility |
|---|---|---|---|---|---|
| (S♀×L♂) ♂backcrossS♀ (n = 5) | 3.80 ± 2.95a | 3.40 ± 3.88b | 77.50 ± 43.66a | 0.47 ± 0.35a | 0.980 |
| (S♀×L♂) ♀backcrossL♂ (n = 3) | 4.33 ± 1.53a | 4.00 ± 1.00ab | 94.40 ± 9.62a | 0.59 ± 0.08a | 1.230 |
| (S♂×L♀) ♂backcrossL♀ (n = 3) | 3.33 ± 1.53a | 3.00 ± 2.00b | 83.30 ± 28.87a | 0.76 ± 0.21a | 1.382 |
| (S♂×L♀) ♀backcrossS♂ (n = 3) | 6.67 ± 2.08a | 6.33 ± 2.52ab | 93.30 ± 11.55a | 0.61 ± 0.10a | 1.110 |
| S♂×S♀ (n = 15) | 7.00 ± 4.95a | 6.37 ± 1.69ab | 91.00 ± 13.98a | 0.48 ± 0.33a | —— |
| L♂×L♀ (n = 18) | 8.67 ± 2.89a | 7.29 ± 2.39a | 84.08 ± 30.29a | 0.55 ± 0.34a | —— |
| | df = 5; F = 1.488 P = 0.249 | df = 5; F = 1.829 P = 0.164 | df = 5; F = 0.386 P = 0.851 | df = 5; F = 0.615 P = 0.690 | —— |

Data in the same column are mean ± SD, different letters indicate significant differences (*P*<0.05). L: Liaoning clade; S: Shandong clade. n represents the number of replications.

crosses strongly suggest there is no mating incompatibility between these populations, and that there is likely significant gene exchange between two clades [29, 30]. Indeed, previous studies have documented such gene flow between these populations, which is what would be expected, given the small geographical distance between the two clades' distributions [12]. Generally speaking, the formation of species mainly includes the allopatric speciation and the sympatric speciation. Allopatric speciation is also called geographical speciation, which is considered by modern evolutionists to be the main way of population speciation into reproductive isolation species [31–33]. Before speciation, the environment is relatively uniform, the populations exist with a single or a series of similar form. Then, there are environmental differences and subgroups that adapt to each community. However, there was no reproductive isolation.

In this paper, we examined samples from two populations of *A. mali*, Qingdao population of the Shandong clade and the Qinhuangdao population of the Liaoning clade. We found no reproductive isolation between these two populations. However, populations from more widely separated western or southern locations (e.g., in Xinjiang or Yunnan Provinces) of these clades may be more reproductively isolated and this possibility needs further clarification.

## Supporting information

**S1 Fig. The linear arrangement of mitochondrial genomes in the sequences of two clades.** (DOCX)

**S1 Table. Primers used for amplification of the mitochondrial genome.** (DOCX)

**S2 Table. Comparisons of the mitochondrial genome between two clades of *Aphelinus mali* in China.** (DOCX)

**S3 Table. The base content, AT skewness and GC skewness of the two *Aphelinus mali* clades compared to other Hymenoptera.** (DOCX)

**S4 Table. The base composition of mitochondrial genomic coding gene and rRNA gene of two clades of *Aphelinus mali* in China.** (DOCX)

## Acknowledgments

The authors are highly thankful to Roy Van Driesche for English and Scientific editing as well for suggestions and advice on this research.

## Author Contributions

**Writing – original draft:** Lingjun Du, Jianing Yu, Mengyu Chi.

**Writing – review & editing:** Min Su, Muhammad Yasir Ali, Ziwen Teng, Yinjun Fan, Xiumei Tan, Hongxu Zhou.

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
