## [Decision Letter · Decision Letter 0]

11 Oct 2021

PONE-D-21-21080Differences in morphology, mitochondrial genomes, and reproductive compatibility between two clades of Aphelinus mali (Hemiptera: Aphididae) in ChinaPLOS ONE

Dear Dr. Zhou,

Thank you for submitting your manuscript to PLOS ONE. After careful consideration, we feel that it has merit but does not fully meet PLOS ONE’s publication criteria as it currently stands. Therefore, we invite you to submit a revised version of the manuscript that addresses the points raised during the review process.

We look forward to receiving your revised manuscript.

Kind regards,

Bi-Song Yue, Ph.D

Academic Editor

PLOS ONE

Journal Requirements:

2. In your Methods section, please provide additional location information of the collection site, including geographic coordinates for the data set if available.

3. In your Methods section, please provide additional information regarding the permits you obtained for the work. Please ensure you have included the full name of the authority that approved the collection site access and, if no permits were required, a brief statement explaining why.

National Natural Science Foundation（31772232) .

 Taishan Mountain Scholar Constructive Engineering Foundation of Shandong, China

Reviewers' comments:

Reviewer's Responses to Questions

**Comments to the Author**

1. Is the manuscript technically sound, and do the data support the conclusions?

Reviewer #1: Partly

Reviewer #2: Yes

Reviewer #3: Partly

Reviewer #4: Yes

Reviewer #5: Partly

2. Has the statistical analysis been performed appropriately and rigorously? 

Reviewer #1: Yes

Reviewer #2: Yes

Reviewer #3: Yes

Reviewer #4: I Don't Know

Reviewer #5: Yes

3. Have the authors made all data underlying the findings in their manuscript fully available?

Reviewer #1: No

Reviewer #2: Yes

Reviewer #3: No

Reviewer #4: Yes

Reviewer #5: Yes

4. Is the manuscript presented in an intelligible fashion and written in standard English?

Reviewer #1: Yes

Reviewer #2: No

Reviewer #3: Yes

Reviewer #4: No

Reviewer #5: No

5. Review Comments to the Author

Reviewer #1: The manuscript entitled “Differences in morphology, mitochondrial genomes, and reproductive compatibility between two clades of Aphelinus mali (Hemiptera: Aphididae) in China” by Su et al. compared and analyzed the morphological characteristics and the mitochondrial genome of two clades of Aphelinus mali in China, and ran a hybridization experiment between these clades, the results indicate that the historic geographical isolation between these clades has not yet caused reproductive isolation, the populations tested belong to the same species. This study provides useful information for the application of A. mali. However, there are a few clarification need to be made and after revision I suggest the manuscript be acceptable.

There is a fatal problem in the sampling of cross mating experiment. The Liaoning clade was collected in Qinhuangdao, but Qinhuangdao is in Hebei Province, and Hebei Province is connected to both Liaoning and Shandong provinces. The authors need to explain why the population collected in Hebei can represent the Liaoning clade, but not the Shandong clade. Otherwise, the data collected from Hebei Province cannot be used to support the author's views.

In addition, geographic information about Liaoning and Shandong should first be provided in the introduction, so that readers can easily understand whether the two provinces are adjacent or not, and whether there is a natural isolation.

The discussion was not sufficiently developed to fully reflect the value of this study.

There are also some formatting issues, such as missing spaces (Line 98) or extra spaces (Line 85, Line 143, Line 218). Some expressions seem to be improper, and if the manuscript can get further language assistance, it will be more helpful for publication.

Minor suggestions:

Line 51: According to the reference, 20214 should be 2014, please check it. In addition, “bio control” should be modified to “bio-control”.

Line 77: Please check and modify atca25°C.

Line 124: Please check whether it is " colon " or " colony ".

Line 180: "And" at the beginning of a sentence needs to be capitalized, or change the period before "and" to a comma.

Line 181: For the first ")" that appears in this line, the corresponding "(" is not found.

Line 218-219: The latitude information provided here should be provided in the method first.

Line 240: "Since" is not at the beginning of the sentence and should not be capitalized.

Line 254: It is recommended to rewrite the end of this paragraph.

Line 367, 375, 380: The topics of Table 1, Table 5 and Table 6 are too long. The sentences after "Data" could be moved to the bottom of the tables.

Line 371-372: If "SC" and "LC" are abbreviated for the first time, an explanation is required.

Reviewer #2: This manuscript describes the genetic/morphological comparison of two wasp clades that are geographically separated. This study includes hybridization experiments that compliment these data to give strong support for the conclusion that the two clades are indeed still members of the same species. Given that this wasp is an effective biocontrol agent, these results will help inform control strategies. Overall, this study is well planned, properly executed and the data are appropriately interpreted.

One very important error is that the title contains the wrong Order and Family of the wasps being studied. It instead refers to the Order and Family of the endoparasitoid’s host.

A related concern: it is not clear to me if the authors have their given/family names in the correct order (given name first, then family name). Please check this.

I would also like to suggest that the authors include a map showing the wasp collection sites to help readers visualize the distances between each. This will assist in the interpretation the results. It would also be helpful to know why different collections sites were used for each of the analyses and not the same sites. Could this have impacted the data?

Line 124 implies that the authors established a colony of aphids to use as a host for the wasps. If this is true, please include more information on how this colony was maintained.

Tables 5 and 6: The descriptions for each table indicate that statistical analyses were performed for the rows, but I believe the analyses would have been performed for the columns (since, for example, the number of pupae between crosses would have been compared and not the # pupae, adult progeny, % emergence, etc. for the same cross). Please check this.

The References section is incomplete. I noticed that Su et al 2017 (mentioned in the text, which describes an important companion study) is not in this section. I did not take the time to see if this is true of other citations, but I would ask the authors to ensure all citations are in this section. Also H.X. Zhou et al 2017 is missing the first three authors’ names. Additionally, I believe the authors did not follow the journal’s formatting guidelines for this section. I would ask the authors to check this entire section for errors.

I have also lightly edited the text in a few areas and made some suggested changes for clarity and/or grammatical correctness. I have uploaded the pdf for the authors to view.

Reviewer #3: This work compared morphology, mitochondrial genomes, and reproductive compatibility of two clades of Aphelinus mali from two provinces in China. The data is interesting, but I think the scientific importance of this research is limited due to narrow sample collection. The author only concentrated on the population from Liaoning and Shandong, more important, the populations of A. mali in these two locations might be originated from abroad Japan (Liaoning clade) and former Soviet Union (Shandong clade) according to your introduction. The other major concern is that the author use the clades from different locations in different types of experiments. For example, the Taian population of the Shandong clade and the Dalian population of the Liaoning clade were used in morphological determination; the Taian population of the Shandong clade and the Huludao population of the Liaoning clade were used in mitochondrial genome analysis; the Qingdao population of the Shandong clade and the Qinhuangdao population of the Liaoning clade were used in crossing mating test. The varied-originated samples (although in same province) would weaken the conclusion made by the authors, that is they supposed that these two clades belong to the same species. Moreover, Qinhuangdao located at Hubei province rather than Liaoning. Additionally, the methodology in this experiment is not well explained, nor is sufficient context provided. Many punctuation misuse were occurred in the context.

Finally, below I make a number of suggestions that I hope the authors will find useful in improving their manuscript.

Overall, I think the manuscript is not suitable for publication in PLOS ONE at current form.

Abstract

L34 add the space between AT and levels

L34-36 This sentence is confused, I think it better should be improved.

L41,42 I suggest to change some key words for the aim with avoiding repetition with the title

Introduction

L45 add space between lanigerum and (Hausmann)

L51 Which publication year is correct?

L51 Change “bio control” to biocontrol or “bio-control”

L52/L54 If the Latin scientific name of a species is placed at the beginning of a sentence, the Latin scientific name should be written in full. Please check throughout the manuscript.

L63 The reference of Su et al. 2017 was missed in the reference list.

L63 Is su et al. 2018 right citation? I found this reference mainly focused on the laboratory comparison of two Aphelinus mali clades from Hebei Province

M&M

L76 The longitude and latitude of collection sites should be listed.

L77 add space between “at” and “ca”, Change petri dishes to Petri dishes.

L78 delete the space behind “daily”

L79 Stored in 20°C? It is a suitable temperature for conservation.

L93-94 The longitude and latitude of collection sites should be listed. It is a very important information for mitochondrial genomic analysis from different locations.

L98 add space between °C and incubator

L101 delete the “.” in the sentence

L104, 106 The sequences of two primers were not provided in the manuscript.

L105-106 16s, 12s,cox1...The letter should better be capitalized. Check throughout the manuscript.

L115 add space after 0.5 uL.

L116-117 What is the sequences of the primer F and R?

L119-120 I have a big doubt, that is, do Qinhuangdao locate at Liaoning Province rather than Hubei Province?

L121-122 I can not understand how do you operate in this step? Why you use a screen cage?

L122-124 The grammar of this sentence is wrong. What is the meaning of “colon”?

L127 If the sample was collected from Qinhuangdao, I did not think it is Liaoning population.

L127-128 How many insects did you test in each repeatation during the cross mating experiments? Were the collected parasitoids used in cross mating test once they gathered from the field?

143 The space should be delete.

L143-145 What ANOVA method did you use?

L148 Full stop was missed in this sentence.

Results

L155-159 I suggest to reconstruct this paragraph, which is highly similar with abstract. And P, F, df value were not listed in this part.

L160-188 Did you submit the mitochondrial genome sequence of these two population of A. mali to NCBI? What are the accession No.?

L165-171 What are the accession No. of other Hymenoptera insects mentioned in your manuscript? Beside the accession No., the collection sites, reference source etc. of other Hymenoptera insects should be offered in this manuscript, they are very important. Additionally, the method description of this part was not found in M&M part.

L173 I suggest the gene name such as atp8 should be capital. Please check throughout the manuscript.

L180 Change and to And

L181 delete “)”

L189 “females of A.mali that had not mated” changed to unmated A. mali females

L190-191 This conclusion is contrary to what you observe.

L191 add full stop.

L192-200 The values of t P df were missed.

Discussion

L203-204 So, the mitochondrial genomes structure of these two clades were incompleted?

L207 “fertileF1female” Please add space

L216 because of influences due to photoperiod, Change to “because of photoperiod influences”

L218 delete unnecessary space

L218-219 (Liaoning clade: 11 122°31'E, 39.20°N, Shandong clade: 116°41'E, 39°91'N). The latitude of these two locations was very close.

L220-223 However, you did not conduct the phylogenetic analysis in your research. Thus, I suggest to delete this paragraph.

L228 What is the general range of skew for wasps?

L230 I think this should be revised. How can you ensure that there have no significant difference?

In fact, the A+T content of these two clades were within the range of other Hymenoptera insects.

L238 Publication year missing.

L239 Change”,” to “.”

L243-244 How can you interpret that significant gene exchange were occurred between two clades, only by the crossing test?

L255-257 in cross mating test, I think this should be emphasized, because you use the clade from different location in different types of experiment. For example, the Taian population of the Shandong clade and the Dalian population of the Liaoning clade were used in morphological determination

Reference

Reference should be carefully checked. Many reference did not follow the criterion of this journal.

Table 3

As mentioned before, many necessary information was missed.

Pergacondei sp. Or spp.?

Table 5

S♂×♀? L♂×♀?

What is # represent for?

Reviewer #4: The manuscript PONE-D-21-21080, “Differences in morphology, mitochondrial genomes, and reproductive compatibility between two clades of Aphelinus mali (Hemiptera: Aphididae) in China” by Min et al. attempted to determine the variations between geographically distinct clades of A. mali parasitoid wasps that established in two provinces in China (Liaoning and Shandong). The endoparasitoid Aphelinus mali is a natural enemy of aphid Eriosoma lanigerum, which is a quarantine pest of apples, thus exploited in biocontrol measures. It is therefore crucial to conduct more studies on these parasitoid wasps to reduce agricultural losses.

While the authors are applauded for their efforts in this work, and the claims they make out of it, showing that the geographical of the two clades has not resulted in reproductive isolation, this manuscript requires the following considerable revisions.

I have major reservations especially under the entire methodology section of the manuscript, which lacks clarity, with several subsections containing wrong and misleading information. This section (methods) should be revised. It is worth noting that the earlier sections of the manuscript (Abstract and Introduction) are well written and in Standard English with good flow of information, but this trend decreases afterwards. Specific concerns are highlighted below.

Title

1. The title: “Differences in morphology, mitochondrial genomes, and reproductive compatibility between two clades of Aphelinus mali (Hemiptera: Aphididae) in China”

The authors have interchangeably described the parasitoid to belong to Orders: Hemiptera and Hymenoptera in other parts of the text. The correct Order classification is Hymenoptera and not the former.

2. I suggest insertion of the common name of Aphelinus mali in the title, such as ‘wooly aphid parasite’; or simply – ‘parasitoid wasp’ (or parasitic wasps); i.e. “Differences in morphology, mitochondrial genomes, and reproductive compatibility between two clades of parasitoid wasp Aphelinus mali (Hymenoptera: Aphididae) in China”

Abstract

3. Lines 25 – 28: In order to clarify the genetic relationship between them, we compared and analyzed the morphological characteristics and the mitochondrial genome of each, and ran a hybridization experiment between these clades. Revise the statement to; …..between these two clades, we……, and performed a hybridization experiment.

4. Line 32, 34: ………was14415bp.., …ATlevels… Spacing errors have been observed through out the text. Correct these errors – please also check line 44, line 49 (citation), line 52, line 85, line 98, et cetera.

Introduction

5. Lines 47 - 48: In recent years, the damage caused by this aphid has become a serious problem in China, and the pest’s range has continued to increase … Quantify the extent of the problem for the reader to appreciate the burden of this pest.

6. The referencing format used by the authors does not conform to the PLoS ONE guidelines. The reference in line 51 … (Zhou et al. 20214), there is a typo for the year

7. Lines 58 – 64: long sentence, consider splitting. Another long sentence, which requires attention appears under M & M section, see lines 79 – 83.

8. Lines 68: A. mali is many times correctly spaced (between the genus and species name), other times it is not (i.e. Line 68, 71). Maintain consistency.

9. Line 71: “We also ran…” rewrite this sentence.

Methods

10. Lines 75-76: Indicate the GPS coordinates of the aphids field sampling and/or map them. It is not clear for reader to deduce the sampling strategy used by the authors. Again, how many times were the field collections performed.

11. Line 77: Guide the readers how the parasitized aphids were identified. “….. and held atca25°C…” correct this typo

12. Line 78: At what age post emergence? same age? or mixed older and young. Specify the exact age were parasitoids used for the experiments

13. Line 86: “This system was used to capture photographs…” This part of the sentence is redundant. Rewrite, or merge with the preceding sentence.

14. Line 91: Were any replicates conducted for the morphological measurements?

Please provide the criteria followed to choose a sample size of 100 A. mali (50M and 50F) from each sample area?

15. Line 93: “Mitochondrial genomic sequencing” write correctly as “Mitochondrial genome sequencing”

16. Major reservation under subsection “Mitochondrial genome sequencing” (Lines 93 – 117): It is unclear how the mitochondrial genome sequencing was performed in this study. Rewrite this subsection for clarity and smooth flow of information and avoid making unclear statements. First, the sequence of events is confusing, line 101: The sending out of DNA samples to Wuhan Transduction Bio Co., LTD for sequencing. This should come after DNA extractions, or were the parasitoid wasp samples sent to the company? Again, Line 102 – 103: says “For DNA extraction, we used the Takara Genomic extraction Kit, specific Primer LA Taq Polymerase, MgCl2 solution and pMD18-T Vector” It is disturbing to note that the authors seem not to understand basic processes (such as DNA isolation, PCR, gene cloning, sequencing), as shown by mixed up protocols. Rewrite these methods accurately. Furthermore, to avoid confusing the reader, clarify the specific study activities conducted at the Wuhan Transduction Bio Co.

17. Line 103-104: “For sequencing we used the following procedure.” Rewrite for clarity. What you describe next is not sequencing, but gene amplification (or PCR) process.

18. Line 109: delete “specific process” that comes after “35 cycles”

19. “The PCR product was then cloned into a PMD-18T (Takara, JAP) vector or sequenced directly with PCR product in…” This sentence is not entirely accurate. Please rewrite.

20. Line 112: “The sequencing results were then spliced and assembled…” use correct term/s instead of “sequencing results”.

21. Line 116-117: Use the correct symbol for primer micromolar (i.e. μM)concentrations, instead of micrometers (μm).

22. Line 121-124: Rewrite these sentences. Avoid contraction such as “doesn't” replace “…making sure…” Also note the mistake in “colon” instead of colony.

23. Line 125: Crosses and reciprocal crosses between the ….. Revise as it is not clear which crosses are been referred for the former

24. Line 128: … write ‘diameter’ in full, instead of “diam”

25. Line 129: “…16:8 L:D…” write as “…16:8 h L:D…”

26. Line 131: Delete “her”

27. Line 135: Check spelling mistake in “parasitoid…”

28. Lines 138 – 139: Redundancy for the statement 136-7. Delete

29. The authors carried out cross mating experiments for reproductive compatibility. However, the fate of the emerged parasitoid adults was not addressed i.e. their longevity and other fitness aspects relative to parental population for conclusive affirmation of claims on reproductive compatibility.

Results

30. The tables provided are not comprehensively described for the data presented and analyses performed.

31. Table 2: Why the redundancy of trnS2 – appearing 4 times; 3 trnS2 with TGA and 1 with GGA Anti/start codons?

32. Table 3: The authors should guide the readers the source of the data presented; for only A. mali clades were sequenced and analyzed. This information should also feature in the methods section.

33. In line 134: The authors state, “We recorded the number of hosts attacked…..”. The dataset for this aspect is not provided.

34. Line 190; 192 all offsprings (plural)

Under this section, it will help if images of Aphelinus mali can be included in this manuscript under morphological descriptions of the two reported clades.

Discussion

35. Lines 203-205: This statement is incomplete and unclear.

36. Lines 220 – 223: Link the statement with the subsequent paragraph for clarity.

37. Lines 228 – 233: There is change of tense; also, this sentence is too long, hence requires splitting.

38. Line 243: ….., and that there likely is significant gene exchange…. Not clear

39. Line 251: The process is that before the….. Use standard English, consider revision.

40. Line 255: “In this paper, we examined two populations of A. mali…” Check again to ensure clarity. My key concern here is the use of the term “populations” instead of ‘samples from two populations’

Reviewer #5: The MS needs major revision

English needs to be improved - many grammatical errors like use of personal pronouns and typographical errors are highlighted in red and blue.

The most important taxonomic character to be considered - length of ovipositor how many times longer than mid tibial length. Usually for larger specimens it is 1.18-1.31x. Body size and length of body parts are not key taxonomic characters for species separation. Also check for colouration of scape pedicel and flagellum as whole.

Care should be taken that only the observed vouchers need to be sequenced (only hind leg or right side legs) remaining body needs to be retained as vouchers.

A brief illustrated diagnosis for A. mali has to be provided with character figure plate for identification.

Whether members of both these clades have exclusive host- E. lanigerum (no other host other than E. lanigerum). Because if both the clade members are not polyphagous then it is not surprising to note that both belong to same species- A. mali. Does larger and smaller clade members infest other known hosts from their respective locations- A mali being a polyphagous and cosmopolitan species!

Are both these localities- Tai'an, Shandong Province and another one in Huludao, Liaoning province geographically isolated by barriers -mountains/water bodies etc. If so then there are very less chances of them to be separate species as they are collected from same host. Whether there is inter-state transportation of apples between both locations- both these answers to questions will give more strength to speciation query.

6. PLOS authors have the option to publish the peer review history of their article (what does this mean?). If published, this will include your full peer review and any attached files.

Reviewer #1: No

Reviewer #2: No

Reviewer #3: No

Reviewer #4: **Yes: **Joel Bargul

Reviewer #5: No

---

## [Author Response · Author response to Decision Letter 0]

17 Nov 2021

The comments that reviewers and editor have been revised, and we submit as an attachment.

---

## [Decision Letter · Decision Letter 1]

28 Apr 2022

PONE-D-21-21080R1Differences in morphology, mitochondrial genomes, and reproductive compatibility between two clades of parasitic wasps Aphelinus mali (Hymenoptera: Aphelindae) in ChinaPLOS ONE

Dear Dr. Zhou,

Thank you for submitting your manuscript to PLOS ONE. After careful consideration, we feel that it has merit but does not fully meet PLOS ONE’s publication criteria as it currently stands. Therefore, we invite you to submit a revised version of the manuscript that addresses the points raised during the review process. Please revise the manuscript to address all the reviewer's comments in a point-by-point response in order to ensure it is meeting the journal's publication criteria. Please note that the revised manuscript will need to undergo further review, we thus cannot at this point anticipate the outcome of the evaluation process.

We look forward to receiving your revised manuscript.

Kind regards,

Miquel Vall-llosera Camps

Senior Editor

PLOS ONE

on behalf of

Daniel Doucet

Academic Editor

PLOS ONE

Journal Requirements:

Reviewers' comments:

Reviewer's Responses to Questions

**Comments to the Author**

1. If the authors have adequately addressed your comments raised in a previous round of review and you feel that this manuscript is now acceptable for publication, you may indicate that here to bypass the “Comments to the Author” section, enter your conflict of interest statement in the “Confidential to Editor” section, and submit your "Accept" recommendation.

Reviewer #1: (No Response)

Reviewer #2: (No Response)

Reviewer #3: (No Response)

Reviewer #4: (No Response)

Reviewer #5: All comments have been addressed

2. Is the manuscript technically sound, and do the data support the conclusions?

Reviewer #1: Yes

Reviewer #2: Yes

Reviewer #3: Partly

Reviewer #4: Partly

Reviewer #5: Yes

3. Has the statistical analysis been performed appropriately and rigorously? 

Reviewer #1: Yes

Reviewer #2: Yes

Reviewer #3: Yes

Reviewer #4: I Don't Know

Reviewer #5: Yes

4. Have the authors made all data underlying the findings in their manuscript fully available?

Reviewer #1: Yes

Reviewer #2: Yes

Reviewer #3: Yes

Reviewer #4: (No Response)

Reviewer #5: Yes

5. Is the manuscript presented in an intelligible fashion and written in standard English?

Reviewer #1: Yes

Reviewer #2: Yes

Reviewer #3: Yes

Reviewer #4: Yes

Reviewer #5: Yes

6. Review Comments to the Author

Reviewer #1: The revised manuscript has been highly improved overall. After further modification, I suggest the manuscript be acceptable.

The question of why the Qinhuangdao population belongs to the Liaoning clade has been explained clearly in Response to Reviewers. But the point of the problem is that the readers are not familiar with the previous research that the authors have done before, so the authors need to clarify this matter in this manuscript, rather than explain it to the reviewers. It is recommended, for instance, to add “(Fig. 1)” at the end of line 127, or to explain in more detail in the manuscript.

There are missing spaces in lines 32, 162, 175, 185 and 388.

Line 48: “WAA” should be defined when it first appears on line 45.

Line 89: please double check it should be “antennal” or “antenna”.

Line 128-133: this paragraph is inconsistent with the tone of the full text, and the narrator is more like a third party who advised the research before the start of the experiment than the experimenter. And, on line 128, it should be “chose” instead of “choose”.

In Fig. 2 Note, it should be “SC”, not “SD”.

Reviewer #2: I want to thank the authors for their careful attention to detail when revising this manuscript. I believe the overall quality of this manuscript has been greatly improved. I have uploaded a PDF with a few minor changes recommended (in the text).

Reviewer #3: After modification, this manuscript has improved to a certain extent, but there are still some important information that has not been modified. In addition, the author did not fully explain the two major concerns I raised last time. This greatly limits the scientific significance of the manuscript. Moreover, the discussion was not well constructed to interpret results obtained in this study. Finally, I think the manuscript is not suitable for publication in PLOS ONE. Several minor suggestion were listed below.

Abstract

L40,41 I suggest to change some key words for the aim with avoiding repetition with the title

Introduction

L49 It should be 5 × 106

M&M

Line 146 revised x to “×”

Results

L170-172 As you mentioned, A+T content of both clades were within the range of other Hymenoptera, rather than having higher values (see your discussion L230-232).

Reviewer #4: Although some key concerns raised previously in my review were addressed, the following comments are not;

Abstract

Line 32: was14415bp not spaced-up

Introduction

The first line of introduction: “Woolly apple aphid (WAA)…”

Lines 46-50: long statement consider splitting

Again, consider spacing up the cited reference numbers at the end statements. Also, Wang et al 2011 is not numbered.

Spacing up of A. mali for many parts of the text not done.

Methods

The coordinates provided should come after the mentioned provinces.

The authors have not addressed the raised comments on sampling strategy and number of sampling times. I feel that the random sampling design is not clear and selection of the sample size of 50 males and 50 females is not well justified.

The mitochondrial genome sequencing protocol is still not again clear, as some of the concerns earlier raised are not addressed. Considering that whole insect samples were send to Wuhan Transduction Bio Co., LTD for sequencing, what activities were done by the authors? What criteria was used for direct sequencing of PCR amplicons or first performing cloning into a plasmid prior to sequencing?

Lines 114: 0.2~1.0 μM instead of 0.2-1.0 μM

Lines 118-119 and 120-121: check repetitions.

Results

The tables provided are not comprehensively described for the data presented and analyses performed. This has not been addressed in the revised manuscript.

In addition, my previous concern on Table 3 (now Table 4: lines 375-377) is not addressed: the authors should guide the readers on the source of the data presented on other Hymenoptera by providing references or source links (only A. mali clades were sequenced and analyzed by this study). This information should also feature in the methods section.

In line 139: The authors state, “We recorded the number of hosts attacked…..”. The dataset for this experiment is not provided.

The map for the sampling sites was provided as suggested, as well as the images of parasitoids, but these two are not cited in the text.

Reviewer #5: All my comments have been answered with justification. The manuscript after inclusion of all reveiwers comments appears to be in an acceptable state.

7. PLOS authors have the option to publish the peer review history of their article (what does this mean?). If published, this will include your full peer review and any attached files.

Reviewer #1: No

Reviewer #2: No

Reviewer #3: No

Reviewer #4: No

Reviewer #5: No

---

## [Author Response · Author response to Decision Letter 1]

9 Jun 2022

Reviewer #1:

1. Why the population collected in Hebei can represent the Liaoning clade, but not the Shandong clade.

We added “(Fig. 1)” at the end of the Cross-mating experiment part, to clarify why the Qinhuangdao population belongs to the Liaoning clade.

2. The other suggestions have been revised.

Reviewer #2:

The suggestions have been revised

Reviewer #3:

1. Abstract

We have changed some key words.

2. Results

Sorry about that, the descriptions of the A+T content were wrong. We have revised "A+T content of both clades were within the range of other Hymenoptera". The AT skew of both clades have higher values.

3. About the sample: 

As mentioned, Aphelinus mali was introduced into China twice: the first introduction was from Japan into Dalian and Lvshun, Liaoning Province (122°31'E, 39.20°N) in 1942, while the second was from the former Soviet Union in 1950 into Qingdao, Shandong Province (116°41'E, 39°91'N). In our previous research, the A.mali in China belong to two clades: Shandong clade and Liaoning clade, and the two clades were very possible concerned with the introduced.

4. The other suggestions have been revised.

Reviewer #4:

Methods:

1. Because of we measured too many indexes of the A. mali, so we should select a large number of samples to be sure the diversity of samples.

2. We collected the insect samples and then sent to Wuhan Transduction Bio Co., LTD for extraction and detection of Genomic DNA, Primer design, LA-PCR amplification and sequencing, and then we analyzed the date.

Results：

1. The references of Table 3 (now Table 4) is [18].

2. "The number of hosts attacked….." is " the number of the pupae in the Table 6. 

3. The map for the sampling sites was cited the reference [11]. The images of parasitoids were the came from our experiment.

4. The other suggestions have been revised.

---

## [Decision Letter · Decision Letter 2]

26 Sep 2022

PONE-D-21-21080R2Differences in morphology, mitochondrial genomes, and reproductive compatibility between two clades of parasitic wasps Aphelinus mali (Hymenoptera: Aphelindae) in ChinaPLOS ONE

Dear Dr. Zhou,

Thank you for submitting your manuscript to PLOS ONE. After careful consideration, we feel that it has merit but does not fully meet PLOS ONE’s publication criteria as it currently stands. Therefore, we invite you to submit a revised version of the manuscript that addresses the points raised during the review process.

Seven reviewers have now provided feedback on your manuscript. Most reviewers highlight the fact that you addressed most of their comments and point to point to some final minor modifications, which I encourage you to address as well. However, Reviewer 7 raises new questions on your approach and your description of the study system. Although these new comments represent a more profound critic of your work in this second round of reviews, I am confident that you should be able to address them thoroughly as well. Please, do so as precisely and clearly as possible, as they do warrant for now the acceptance of your work for publication.

We look forward to receiving your revised manuscript.

Kind regards,

Benjamin Pélissié

Academic Editor

PLOS ONE

Journal Requirements:

**Comments to the Author**

1. If the authors have adequately addressed your comments raised in a previous round of review and you feel that this manuscript is now acceptable for publication, you may indicate that here to bypass the “Comments to the Author” section, enter your conflict of interest statement in the “Confidential to Editor” section, and submit your "Accept" recommendation.

Reviewer #1: (No Response)

Reviewer #5: All comments have been addressed

Reviewer #6: (No Response)

Reviewer #7: (No Response)

2. Is the manuscript technically sound, and do the data support the conclusions?

Reviewer #1: Partly

Reviewer #5: Yes

Reviewer #6: Yes

Reviewer #7: Partly

3. Has the statistical analysis been performed appropriately and rigorously? 

Reviewer #1: Yes

Reviewer #5: Yes

Reviewer #6: Yes

Reviewer #7: N/A

4. Have the authors made all data underlying the findings in their manuscript fully available?

Reviewer #1: Yes

Reviewer #5: Yes

Reviewer #6: Yes

Reviewer #7: Yes

5. Is the manuscript presented in an intelligible fashion and written in standard English?

Reviewer #1: Yes

Reviewer #5: Yes

Reviewer #6: No

Reviewer #7: Yes

6. Review Comments to the Author

Reviewer #1: The authors have revised and clarified most of the previous issues, but there are still some minor revisions to be made and some new issues to be addressed.

Line 64: It is suggested to change "control potential" to biological control potential, or pest control potential.

Line 88: front and hind wings

Line 219: It is suggested to move “[22]” outside the brackets.

Table 2: It is recommended to change to "Forewing, Hindwing" or "Fore wing, Hind wing".

Some italics issues, t (Line 152), F, P (Table 6, Table 7).

There are still some space issues, "239.85±2.87b, 259.67±4.06c, 288.79±4.02a" (Table 2), "84.08± 30.29a" (Table 6), "84.08± 30.29a" (Table 7).

Reviewer #5: (No Response)

Reviewer #6: All my previous comments have been properly addressed. However, I propose more comments when I go through the manuscript:

L48-50: the sentence does not read in a good fashion. It needs to be revised.

L93-95: specify when those samples were collected.

L93-124: this section is too much detailed and I suggest losing weight for this section.

L134-136: how many groups did you set up？As I see it, you shall have at least four groups: Liaoning♀×Shandong♂, Liaoning♂×Shandong♀ as experimental groups, and Shandong♀×Shandong♂, Liaoning♂×Liaoning♀ as control groups. Those key information shall be clearly specified so that readers could get it easily.

Table 1 shall be assigned into supporting information.

It is unnecessary to present all of the table 3 and 4 and 5. Please consider to show only the key results in the main article and the rest could go to supporting information.

Figure 1 and 2 could be shown as subfigures in a large figure.

Is figure 3 an obligatory one to be shown? Otherwise, just remove it or assign it into supporting information.

Despite that the manuscript has been proof-read by a specialist, the readability still needs to be increased. I have seen many sentences that are hard to go through. I suggest authors relying on a native English speaker and/or professional edit service to improve the writing. This is important for the journal which has a wide audience.

Reviewer #7: 1. Aphelinus mali was introduced from Japan and the former Soviet Union into China. The different population sources may divided the parasitoid into two clades. For their morphology, the body size of Liaoning is larger than Shandong as shown in Table 2. Are there other different characters between two clades? Such as body color, wing shade, veins, ect., because at least we know that some species of the genus show obviously different in body color, and some geographic populations may present the same difference.

2. Why do not use high through sequencing? The mitochondrial genome of insect contains 37 genes, including 13 protein-encoded genes, 22 tRNA genes, and 2 rRNA genes. So, the other six tRNA genes must present in the mt genome of the parasitoid, just were not identified. And the mt genomes of chalcids exhibit higher recombination ratio, what about the recombination for the parasitoid?

3. Others need to be corrected:

(1)Eriosoma lanigerum (Hausmann) belongs to the order Hemiptera not Hymenoptera；

(2)“Anterior leg“ should be “Fore leg” in Table 2;

(3)“sp.”should be not Italics in Table 4.

7. PLOS authors have the option to publish the peer review history of their article (what does this mean?). If published, this will include your full peer review and any attached files.

Reviewer #1: No

Reviewer #5: No

Reviewer #6: No

Reviewer #7: No

---

## [Author Response · Author response to Decision Letter 2]

2 Nov 2022

We changed a funder (31371994) to a new one (32172395).

The funder "National Natural Science Foundation (32172395)" play a role in study design, data collection and preparation of the manuscript, the funder " Taishan Mountain Scholar Constructive Engineering Foundation of Shandong, China " play a role in data collection.

Reviewer #1:

The suggestions have been revised.

Reviewer #6:

1.The suggestions have been revised

2. About the writing

The English writing had been improved.

Reviewer #7:

1. Based on the study of the mitochondrial COI gene of this parasitoid, A. mali in China is comprised of two regional clades (named the Shandong and Liaoning clades), we found there were significant difference between two clades of body size, none of body, wing shade and others.

2. Sorry about that, when we did this part of experiment, we chose mitochondrial sequencing first, we will go on the experiment in the future.

3. Others 

We have corrected the mistakes.

---

## [Decision Letter · Decision Letter 3]

13 Dec 2022

Differences in morphology, mitochondrial genomes, and reproductive compatibility between two clades of parasitic wasps Aphelinus mali (Hymenoptera: Aphelindae) in China

PONE-D-21-21080R3

Dear Dr. Zhou,

We’re pleased to inform you that your manuscript has been judged scientifically suitable for publication and will be formally accepted for publication once it meets all outstanding technical requirements. Nevertheless, please address the last formatting points raised reviewers and listed below this message.

Kind regards,

Benjamin Pélissié

Academic Editor

PLOS ONE

Additional Editor Comments (optional):

Reviewers' comments:

Reviewer's Responses to Questions

**Comments to the Author**

1. If the authors have adequately addressed your comments raised in a previous round of review and you feel that this manuscript is now acceptable for publication, you may indicate that here to bypass the “Comments to the Author” section, enter your conflict of interest statement in the “Confidential to Editor” section, and submit your "Accept" recommendation.

Reviewer #1: (No Response)

Reviewer #6: All comments have been addressed

2. Is the manuscript technically sound, and do the data support the conclusions?

Reviewer #1: Yes

Reviewer #6: Yes

3. Has the statistical analysis been performed appropriately and rigorously? 

Reviewer #1: Yes

Reviewer #6: Yes

4. Have the authors made all data underlying the findings in their manuscript fully available?

Reviewer #1: Yes

Reviewer #6: Yes

5. Is the manuscript presented in an intelligible fashion and written in standard English?

Reviewer #1: Yes

Reviewer #6: Yes

6. Review Comments to the Author

Reviewer #1: The revised manuscript has been improved on the whole, although it is regrettable that similar formatting problems appear repeatedly in each revision. I suggest this manuscript to be accepted after a few minor revisions. Please see my comments below (based on the marked edition):

Line 35: "however" needs to be followed by a comma.

Line 82: Add a comma after "Tai’an".

Line 145-146: Please check whether the font size is consistent.

Line 248: If "howere" is "however", it should be followed by a comma.

Line 342: A space is missing before "Sinica".

Line 402: A space is missing before "and".

Reviewer #6: (No Response)

7. PLOS authors have the option to publish the peer review history of their article (what does this mean?). If published, this will include your full peer review and any attached files.

Reviewer #1: No

Reviewer #6: No

---

## [Editor Report · Acceptance letter]

23 Jan 2023

PONE-D-21-21080R3 

Differences in morphology, mitochondrial genomes, and reproductive compatibility between two clades of parasitic wasps *Aphelinus mali* (Hymenoptera: Aphelindae) in China 

Dear Dr. Zhou:

I'm pleased to inform you that your manuscript has been deemed suitable for publication in PLOS ONE. Congratulations! Your manuscript is now with our production department. 

Kind regards, 

on behalf of

Dr. Benjamin Pélissié 

Academic Editor

PLOS ONE